# A clonally expanded nodal T-cell population diagnosed as T-cell lymphoma after CAR-T therapy

Katie Maurer [1,2,3,7], Jackson A. Weir [3,4,7], Adi Nagler[1,2,3],
Nicholas J. Haradhvala [3], Hariharan Bharadwaj [5], Jacob Shapiro[3,6],
Somkene Alakwe[3], Vipin Kumar[3], Brianna Waller[5], Mikaela McDonough[1],
Jamie Dela Cruz[1], Loida Luna[1], Emma Lin[1], Linsey Gong[3], Qiyu Gong[3], Mehdi Borji[3],
Phillip D. Michaels[5], Jacob P. Laubach[1,2], Geraldine Pinkus[5], Gad Getz [3],
Catherine J. Wu[1,2,3,8] ✉, Fei Chen [3,8] ✉ & Caron Jacobson[1,2,8] ✉

Reports of secondary malignancies after chimeric antigen receptor (CAR)-T and possible CAR-T derived malignant transformation necessitate caution. Here we describe a patient with diffuse large B-cell lymphoma who developed new lymphadenopathy 2.5 years after CAR-T in the context of COVID-19 infection with histopathologic features consistent with T-cell lymphoma (TCL). Deep molecular interrogation with genomic sequencing and single-cell spatial transcriptomics reveals a highly proliferative clonal T-cell population co-expressing CD4 and CD8 with biallelic TCR rearrangement and no evidence of the CAR construct. The expanded clonotype displayed T follicular helper (TFH) cell transcriptomic programs and occupies immune-excluded spatial niches within the lymph node, supportive of TFH-like neoplastic T cell behavior. Remarkably, the lymphadenopathy spontaneously resolved on interval imaging. Our data underscore the need for better understanding of post-CAR-T clonal T-cell lymphoproliferative disorders to avoid unnecessary treatment and higher specificity in diagnostic methods for TCL.

The clinical successes of chimeric antigen receptor (CAR) T cells for treatment of lymphoma, leukemia and myeloma have transformed the therapeutic landscape for B-cell malignancies, rendering long-term remissions in patients with few alternatives[1–3]. Despite these clear successes, therapy-related toxicities pose short- and long-term risks for patients following CAR-T, including immune-mediated toxicities after infusion, infections due in part to ongoing B cell aplasia, and secondary malignancies[4].

In November 2023, the FDA issued a warning regarding the risk of T-cell malignancies deriving from the CAR-T manufacturing process[5]. A

low overall risk of secondary malignancy after CAR-T therapy (3–16%) has been reported[6–8]. The majority of these do not appear to result from malignant transformation from CAR-T insertional mutagenesis[7,9,10]. A recent analysis highlighted the overall rarity of these secondary events (6.5%), with very few cases of TCL. One such case was an Epstein-Barr Virus (EBV)-positive TCL arising 54 days after CAR-T cell infusion (axicabtagene ciloleucel [axi-cel]) in a patient with EBV+ diffuse large B-cell lymphoma (DLBCL)[11]. A separate T-cell lymphoma was described following anti-BCMA CAR-T therapy (cilta-cabtagene autoleucel [Cilta-cel]) for multiple myeloma (MM), wherein

[1]Department of Medical Oncology, Dana-Farber Cancer Institute, Boston, MA, USA. [2]Harvard Medical School, Boston, MA, US. [3]Broad Institute of MIT and Harvard, Cambridge, MA, USA. [4]Biological and Biomedical Sciences Program, Harvard University, Cambridge, MA, USA. [5]Department of Pathology, Brigham and Women's Hospital, Boston, MA, USA. [6]Biophysics Program, Harvard University, Boston, MA, USA. [7]These authors contributed equally: Katie Maurer, Jackson A. Weir. [8]These authors jointly supervised this work: Catherine J. Wu, Fei Chen, Caron Jacobson. ✉e-mail: catherine_wu@dfci.harvard.edu; chenf@broadinstitute.org; caron_jacobson@dfci.harvard.edu

the CAR transgene had integrated into the *PBX2* gene[12]. Still another indolent CD4+ CAR-derived TCL in the small intestine was recently reported, wherein whole-genome sequencing (WGS) identified a lentiviral insertion site in the *SSU72* gene[13]. A more recent report described an aggressive CAR+ peripheral T cell lymphoma arising after treatment with tisagenlecleucel for treatment of primary central nervous system lymphoma, which also harbored mutations in *TET2* and *DNMT3A*, suggesting clonal hematopoiesis had contributed to its development[14].

Other case reports have noted unexpected effects on cellular function as a result of CAR-T integration in unusual contexts. In one instance, CAR-T insertion into a single leukemic cell promoted early relapse and therapeutic resistance in a patient with B-acute lymphoblastic leukemia (ALL)[15]. In another, exuberant proliferation and activity of a CAR-T cell in which the CAR transgene had disrupted the *TET2* gene conferred enhanced anti-tumor function[16]. While the benefits offered by CAR-T therapy in inducing enduring remissions in a substantial proportion of patients are largely felt to outweigh these potential long-term risks associated with treatment[17], further work is needed to better characterize these cases of post-CAR-T suspected malignancies.

In this work, we analyze our clinical cohort of patients treated with CAR-T who later presented with possible TCL. We identify three potential cases of TCL, one of which we further characterize by WGS and spatial transcriptomic and T cell receptor (TCR) sequencing to identify putative drivers of proliferation.

## Results

### Clinical cases of possible post-CAR-T lymphoma

We systematically assessed our clinical database of patients with non-Hodgkin lymphoma (NHL), MM, and B-cell ALL receiving one of the six FDA-approved CAR-T products to quantify the incidence of secondary malignancy at our center from 2017 to 2023 with a focus on possible T-cell lymphomas. From 626 total patients, we identified 43 cases of secondary malignancy (6.9%, $n = 12$ myelodysplastic syndrome [MDS]/ acute myeloid leukemia [AML], $n = 28$ solid tumors, $n = 3$ possible TCL); six cases were in MM patients and 37 in NHL (Fig. 1). Of the

possible TCLs, one was a bona-fide case bearing the CAR transgene after Cilta-cel, as described above[13]. Of the remaining two cases, one occurred in a patient treated with Cilta-cel for MM who presented with an 8 mm facial lesion approximately 6 months after CAR-T. Biopsy revealed a population of lymphocytes positive for CD2, CD5, CD3, CD8, MUM-1, TCR-BF1, Granzyme B, and Perforin with Ki-67 of 100%, and negative for CD20, CD138, CD30, CD4, EBV, and TCR-Delta. This was diagnosed as Stage I peripheral TCL. Remarkably, the lesion spontaneously resolved after biopsy and the patient received no TCL therapy.

The second case was an 80-year-old man treated with third-line Axi-cel for transformed follicular lymphoma (FL). The full clinical course leading up to CAR-T is described in Fig. 2a (also see "Methods"). The patient achieved a complete metabolic response on the first restaging PET/CT 30 days after CAR-T, which continued for 2.5 years. Two years after his CAR-T cell infusion, the patient developed COVID-19 infection (despite 6 total vaccinations) with a protracted course, complicated by recurrent superimposed bacterial pneumonias. In the midst of these infections, he presented with weight loss and fatigue, prompting a PET/CT scan, which revealed a highly FDG-avid enlarged right cervical lymph node (LN), concerning for disease relapse or new malignancy (Fig. 2b).

Hematoxylin and eosin (H&E) stain of a core needle biopsy demonstrated architecture effaced by an infiltrate of medium to large cells with irregular/folded nuclei, condensed chromatin, variably prominent nucleoli, and scant eosinophilic cytoplasm with small, scattered lymphocytes in the background (Fig. 2c). By immunohistochemical staining, the abnormal appearing cells expressed mature T-cell markers (CD3, CD2, CD5, CD7), with many positive for both CD4 and CD8, and were positive for TCR-BF1. Scattered cells were also positive for MUM-1 as well as BCL-6, Stathmin, PD-1, and BCL-2 suggestive of a TFH subtype. The abnormal lymphoid cells were negative for CD10, CD30, ALK-1, CD20, PAX5, CD79a, CD21, and C-MYC. Only rare cells stained positive for TCR-Delta. By Ki-67 staining, the cellular proliferation rate was 40–50%, and EBV in situ hybridization was negative. Altogether, the morphologic and immunophenotypic profile was thought to be most in keeping with a TFH TCL.

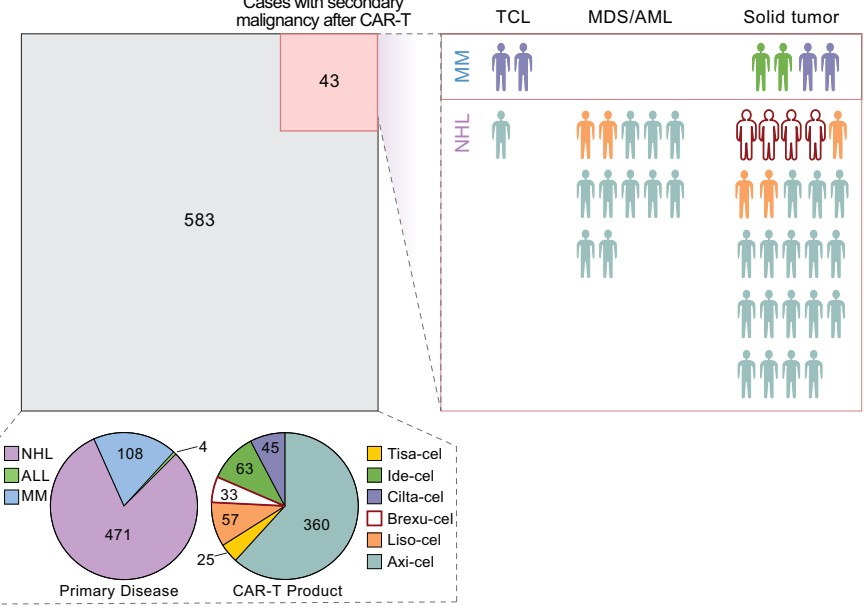

**Fig. 1 | Schematic of entire clinical CAR-T cohort treated at DFCI from 2017 to 2023.** 583 patients did not develop secondary malignancy after CAR-T (gray box). Left pie chart–breakdown of primary disease of these 583 patients. Right pie chart–breakdown of CAR-T product received by these 583 patients. Red box–43 patients developed secondary malignancy after CAR-T, with 3 TCL cases, 12 myelodysplastic syndrome (MDS)/acute myeloid leukemia (AML), 28 solid tumor. Colors indicate CAR-T product each patient received.

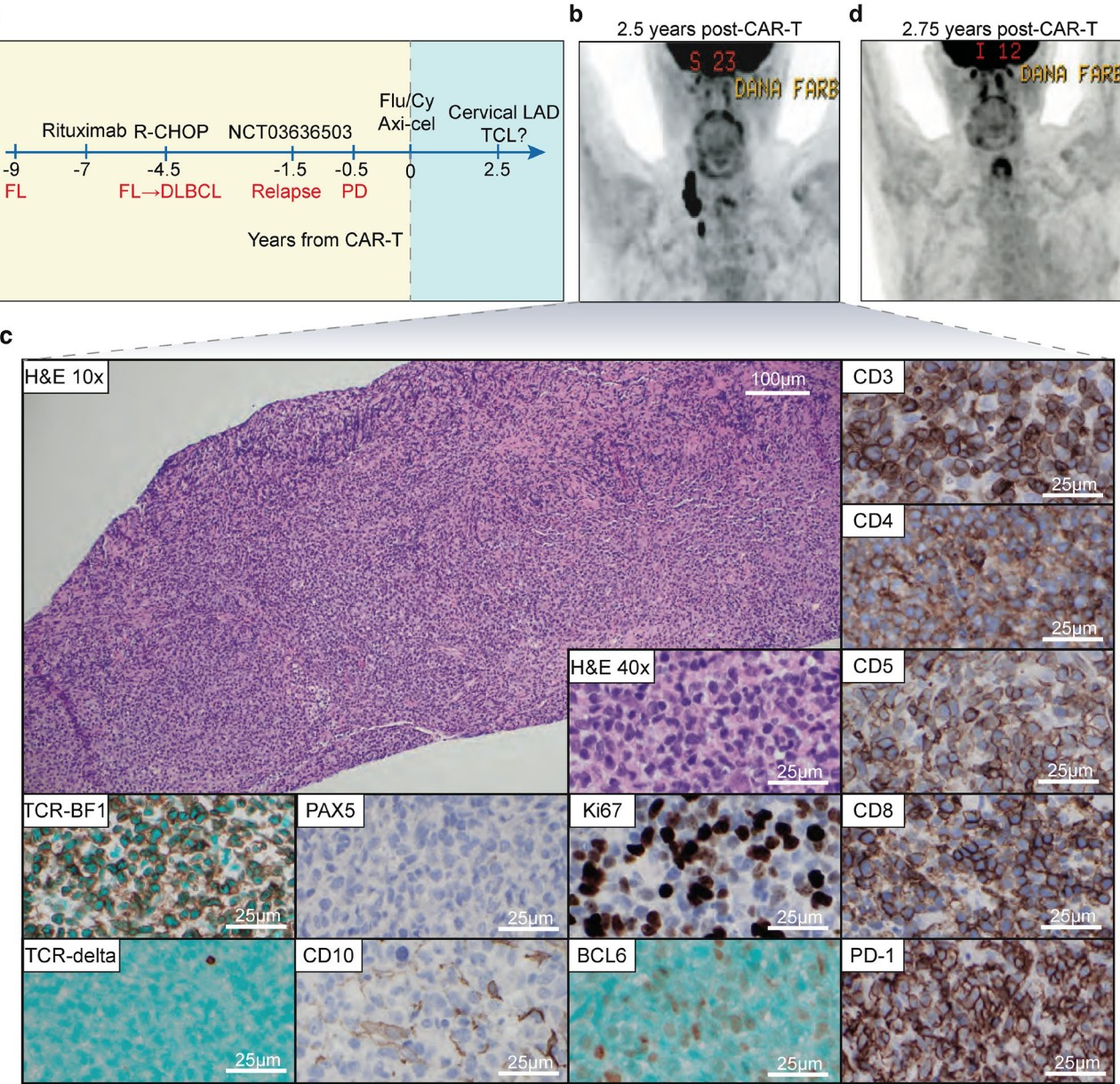

**Fig. 2 | Clinical timeline and lymph node biopsy. a** The patient was originally diagnosed with stage III, grade 1–2 follicular lymphoma (FL) 9 years prior to CAR-T. Upon progression, he was treated with rituximab and achieved a partial response (PR) for 2.5 years before he relapsed with transformed disease. He was then treated with rituximab, cyclophosphamide, doxorubicin, vincristine, and prednisone (R-CHOP) for 6 cycles with a complete response (CR) lasting 3 years. He was treated on a clinical trial (NCT03636503) with rituximab/avelumab/utomilumab combination immunotherapy, with a PR followed by progression after 6 months. He then received CAR-T therapy as third-line therapy. Yellow shading indicates time prior to CAR-T, blue shading indicates time post-CAR-T. **b** PET/CT image of the new lymphadenopathy presenting 2.5 years after CAR-T. **c** Representative immunohistochemistry images of T- and B-cell markers of the from a single core lymph node biopsy. **d** PET/CT image of spontaneous regression of the lymphadenopathy 6 weeks after initial scan.

Moreover, TCR gene rearrangement studies demonstrated a clonal process with a Vγ1-8 primer, supporting a diagnosis of malignancy. Clinical flow cytometry demonstrated sample viability of <5%, limiting interpretation. Given the patient's history, the highest concern was for TCL arising from a CAR-T cell. Although the biopsy was diagnostic for TCL, the clinical scenario of lymphadenopathy arising in the context of COVID-19 infection and bacterial pneumonia raised the possibility that this represented an infectious or inflammatory phenomenon, and the patient had no other signs or symptoms of progressive malignancy. Thus, the decision was made to observe with short-interval repeat imaging. Indeed, PET/CT 6 weeks later demonstrated spontaneous near-complete resolution of the cervical lymphadenopathy with complete resolution by 3 months (Fig. 2d).

## Genomic characterization of possible TCL

To determine the cellular etiology and molecular characteristics of this aberrant T-cell population, remaining cervical LN formalin-fixed paraffin embedded (FFPE) sections and genomic DNA derived from peripheral blood mononuclear cells (PBMCs) obtained 12 months post-CAR-T ("normal") were evaluated by WGS. To assess for the presence of the CAR construct, we aligned to a custom reference genome including the axi-cel sequence[18,19] ("Methods"), and identified zero reads mapping to the axi-cel construct with high mapping confidence (MAPQ ≥ 30). We estimate the sequencing depth provides 95% power to detect reads mapping to the scFv region of the CAR construct (which is distinctly mappable from the human genome) even if the CAR were present in as few as ~3.4% of cells ("Methods"). Thus

observing no mapping reads indicates that the clonal T-cell population was not derived from a CAR-T cell. Copy number analyses revealed no large-scale aberrations, but focal deletions at the TCR alpha, beta, and gamma loci were estimated to be present in 67% of cells by ABSOLUTE. MiXCR[20] identified two rearranged clonotypes at each of the three loci (Supplementary Data 1).

The spontaneous resolution of lymphadenopathy raised the possibility that this clonal population did not represent TCL, despite the aberrant histologic appearance and phenotype (e.g., CD4+ CD8+ double positive), apparent clonality, and presence of TFH markers, the combination of which are considered diagnostic of TCL[21]. Of 4584 total mutations (19 nonsilent variants, Supplementary Data 2) that were LN-specific (i.e., not in normal PBMCs), one was identified in *TET2*, which was a frameshift mutation found in exon 3 resulting in a premature stop codon. While not seen in our normal sample, which was collected at 1-year post-CAR-T, this *TET2* mutation was found in the clinical next-generation sequencing panel performed on PBMCs collected three months after the new lymphadenopathy presented, raising the possibility of clonal evolution or clonal hematopoiesis of indeterminate potential. Nevertheless, at such a high allele frequency, it is likely that the mutation is present in the atypical T cells in addition to the myeloid compartment and could be a driver of abnormal T cell proliferation. Of the other 18 non-silent mutations, none have been previously linked to the development of TCL or T cell lymphoproliferative neoplasms. A point mutation was observed in *HIP3K*, a kinase that can bind to Fas, leading to phosphorylation of Fas-associated death domain (FADD), which plays a key role in T cell apoptosis and proliferation[22,23]. While *HIP3K* mutations have not been specifically linked to TCLs, dysregulation of FADD phosphorylation has been implicated in the proliferation capacity of T cell lymphoblastic lymphoma[24] and could be a potential mutation of interest for future study.

## Spatial transcriptomic interrogation of nodal pathology

To more deeply interrogate the clonality, phenotype, and spatial microenvironment of this hyperproliferative T-cell population, we performed single-nuclei spatial transcriptomics on a fresh-frozen core needle biopsy of the suspicious LN with Slide-tags[25]. Previously, Slide-tags enabled spatial profiling of nuclei with 3' capture chemistry of mRNA transcripts. However, 3' capture chemistry poses challenges for TCR sequencing since the variable region is at the 5' end of the TCR mRNA transcript. For improved recovery of TCR sequences, we adapted Slide-tags to be compatible with 5' capture chemistry ("Methods"). Slide-tags 5' snRNA-seq revealed a LN composition dominated by transcriptionally abnormal *CD4+ CD8+* double positive T-cells, making up 51% of profiled nuclei (Fig. 3a). The rest of the LN was composed of 20% macrophages, 10% regulatory T-cells, 5% fibroblasts, 5% *CD4+* T-cells, and smaller proportions of *CD8+* T-cells, NK cells, endothelial cells, and dendritic cells. No B cells were detected. The *CD4+ CD8+* T-cells were distributed across the full area of the profiled LN in a spatially disorganized manner, uncharacteristic of normal LN architecture (Fig. 3b). To examine these T-cell phenotypes, we quantified the usage of T-cell gene expression programs defined from a diverse collection of healthy, COVID-19, cancer, and autoimmunity patient samples (Fig. 3c and Supplementary Data 3)[26]. The double positive T-cells scored highly for TFH and T peripheral helper cell program usage. Differential gene expression analysis between the double positive T cell population and all other profiled T cells revealed an enrichment for genes involved in proliferation, hypoxia, and stress response, further supporting an abnormal cellular phenotype (Fig. 3d). Concordant with the WGS analysis, we mapped the snRNA-seq results to a custom reference containing the axi-cel construct and did not detect any spatially mapped nuclei containing at least one count of an axi-cel mapped read ("Methods"). Additionally, qPCR to detect the CAR construct sequence in the Slide-tags library from the biopsy was negative for transgene amplification (Supplementary Data 4).

The majority of *CD4* or *CD8* single positive T-cells belonged to single member or small-sized clonotypes (Fig. 3e). In marked contrast, the *CD4+ CD8+* T-cells belonged to a highly expanded clonotype, consisting of two alpha chains (CASPGGLTGGGNKLTF, CALSHPFRNSGNTPLVF) and two beta chains (CASSLVVWGRGL-NEQFF, CASSQQDSRNTIYF), suggestive of an atypical biallelic rearrangement of both alpha and beta chains (Supplementary Data 5)[27–29]. Examining public CDR3α and CDR3β TCR clonotypes, we identified 6 alpha chain TCRs and 2 beta chain TCRs with reported specificity to viral antigens (Supplementary Data 6 and "Methods"), but the highly expanded clonotype did not match any known viral-reactive TCRs. Double positive T-cells scored lowly on a transcriptional signature for viral antigen reactivity, despite the lymphadenopathy arising during COVID-19 infection (Fig. 3f). The highly clonal nature of the aberrant *CD4+CD8+* double positive population supports the notion of TCL, though may also be consistent with a non-malignant lymphoproliferative process.

Finally, we leveraged the spatial data to assess the immune microenvironment surrounding the expanded double positive T-cell population. While double positive T cells spatially clustered with each other, their neighborhoods were depleted of DCs, *CD8+* T cells, *CD4+* Treg cells, endothelial cells, and fibroblasts (Fig. 3g). We observed a weak enrichment for macrophages and pDCs in the neighborhoods of double positive T cells. Overall, the aberrant T cell population exists in immune-excluded spatial niches.

## Discussion

In the current case, pathology review seemingly confirmed a diagnosis of TFH TCL based on conventional immunohistochemical definitions and T-cell clonality. Thus, immediate treatment with TCL-directed therapy would have been reasonable, given the presumptive diagnosis of post-CAR-T TCL. Yet the spontaneous resolution of the lesion, arising in the context of COVID-19 infection and superimposed bacterial pneumonia, argues against this and raises the alternative possibility of a reactive nodal process, where initiation of treatment would have been unnecessary. Alternatively, this case may have been a true TCL with an unusual indolent course that spontaneously remitted in a post-infectious setting. Our understanding of post-CAR-T lymphoproliferative processes is evolving as more cases arise, and conventional pathologic diagnosis of TCL may overestimate the true burden of aggressive post-CAR-T lymphomas. This may have the unintended downstream effect of preventing patients from having access to this potentially curative therapy due to referral biases stemming from concerns about rates of secondary malignancy. Further accounting of similar cases with an in-depth study is needed to better define the nature of these lymphoproliferative neoplasms and assess the risks in the context of the established benefits of CAR-T therapy.

This clinical and pathologic discordance further underscores that current diagnostics have inherent limitations, highlighting the need for more precise definitions of malignancy and raising the notion that the clinical course should be carefully considered in diagnosis. One intriguing possibility is that this case could represent a more indolent T cell neoplasm. Though *TET2* mutations have been implicated in T-cell clonal proliferation[16], our WGS and snRNA-seq data suggest these mutations were not unique to the clonal T-cell population. However, given the high allele frequency, it is likely that the *TET2* mutation was present in the aberrant T cell population and contributed to the clonal proliferation and subsequent clinical presentation (Supplementary Fig. 1). Single-cell spatial profiling confirmed one clonotype dominated the biopsied node, displayed a highly proliferative TFH-like transcriptomic phenotype, and occupied immune-excluded spatial niches at the time of biopsy, supportive of neoplastic T cell behavior. Nevertheless, the lesion resolved post-biopsy. Perhaps robust immunosurveillance in the setting of resolution of an infectious process could result in immune-mediated regression of the lesion. Our findings

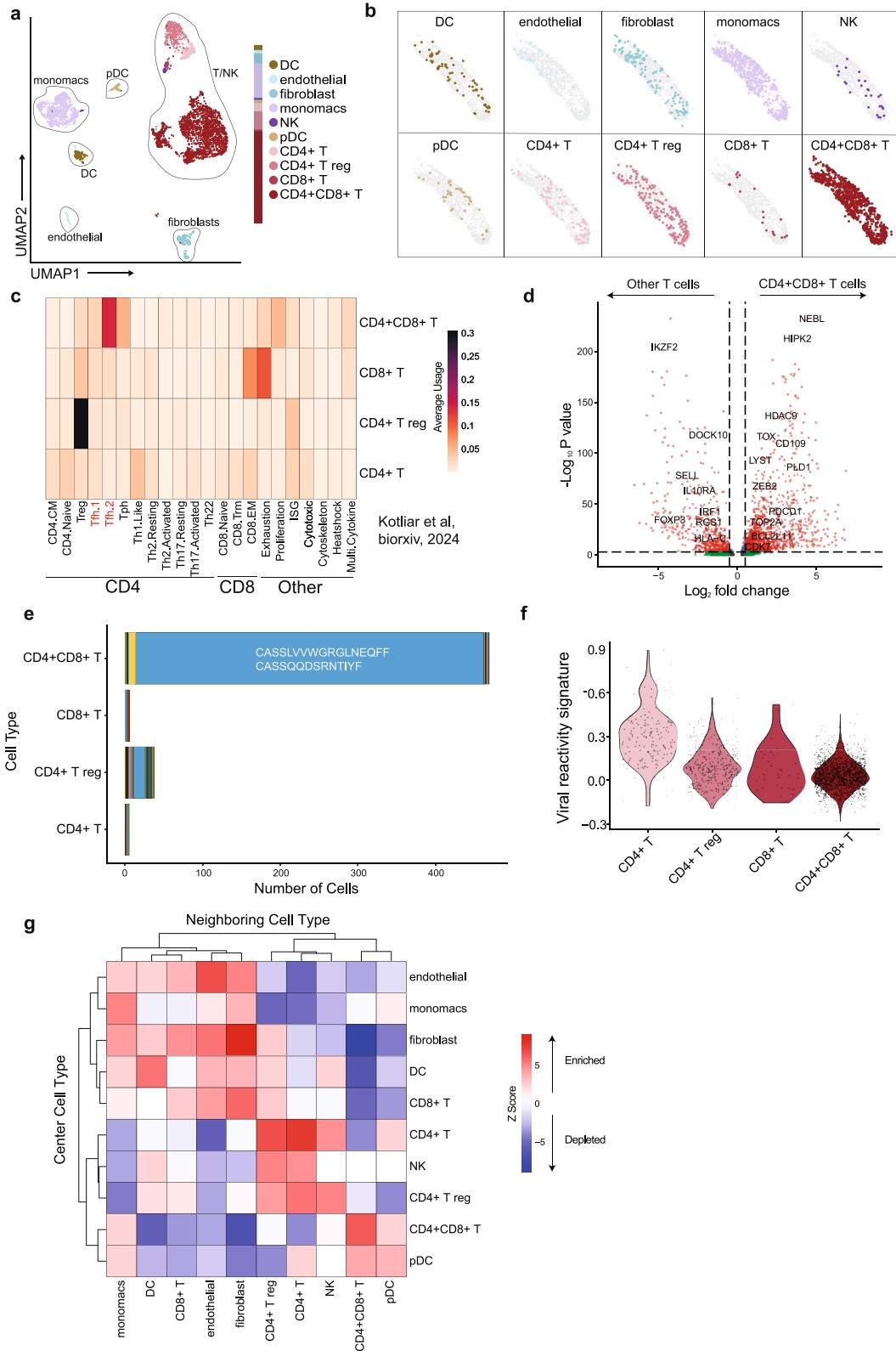

**Fig. 3 | Single-nuclei analysis of lymph node biopsy by 5' Slide-tags. a** UMAP embedding of Slide-tags 5' snRNA-seq transcriptome profiles, colored by cell type assignment. **b** Spatial mapping of snRNA-seq profiles, split by cell type. **c** T-cell gene expression program average usage scores. **d** Volcano plot of differentially expressed genes in the *CD4+ CD8+* T cell population compared with all other T cells using a two-sided Wilcoxon rank-sum test with adjustment for multiple testing using the Bonferroni correction. Select genes are highlighted. **e** Size of beta chain clonotypes by T-cell population. Each color represents a different clonotype. The two beta chain sequences belonging to the most expanded clonotype are written in white text. **f** Module scoring of viral reactivity transcriptional signature by T-cell population. **g** Spatial neighborhood enrichment heatmap where *Z* scores denote the magnitude of enrichment or depletion between a given center cell type and neighbor cell type. *Z* scores are capped at 9 and −9 for visualization purposes.

underscore the need for caution when T-cell lymphoproliferative disorders are identified post-CAR-T and diagnosed as TCL. Current diagnostic criteria may be insufficient when evaluating a patient with a complex immune milieu after CAR-T cell therapy. New diagnostic categories may be needed as we continue to learn more about these post-CAR-T phenomena, particularly in the context of COVID-19 and other viral infections. Careful clinicopathologic correlation is essential when evaluating patients with unusual presentations after CAR-T cell therapy, where clonal lymphoproliferative disorders may instead follow an indolent course without requiring treatment.

## Methods

### Patient characteristics and clinical outcomes

The research was conducted in accordance with the Declaration of Helsinki. Patients gave written informed consent to research and sample banking protocols approved by the Dana-Farber/Harvard Cancer Center Institutional Review Board. For the index patient described above, this patient was originally diagnosed with stage III, grade 1–2 FL 9 years prior to CAR-T therapy. Upon disease transformation to DLBCL, immunohistochemical studies demonstrated that atypical cells were B cells positive for BSAP, CD10, Bcl-2, and Bcl-6. The atypical cells demonstrated scattered c-Myc positivity (40–50% with variable intensity), scattered/focal MUM-1 positivity, and scattered/variable intensity reactivity for CD21. The atypical lymphoid cells were negative for EBV by in situ hybridization (EBER) and for NKX3.1 by immunohistochemistry. Next-generation sequencing was not available on the original lymphoma. He was treated with rituximab, cyclophosphamide, doxorubicin, vincristine, and prednisone (R-CHOP) for six cycles with a complete response (Fig. 2a). His FL relapsed within three years and he was treated on a clinical trial (NCT03636503) with rituximab/avelumab/utomilumab combination immunotherapy, with a PR followed by progression at 6 months. He then received CAR-T therapy as third-line therapy. Five days prior to CAR-T, the patient was treated with lymphodepleting chemotherapy consisting of cyclophosphamide 500 milligrams per meter squared (mg/m$^2$) and fludarabine 30 mg/m$^2$ for 3 days. On day 0, he was infused with $2 \times 10^6$ CAR-T cells per kilogram of body weight. His immediate post-CAR-T course was complicated by cytokine release syndrome (CRS, beginning on day +2 after CAR-T infusion, maximum grade 1 by ASTCT and Lee scoring systems[30,31], treated with steroids and tocilizumab) and neurotoxicity (i.e., immune effector cell-associated neurotoxicity syndrome [ICANS], beginning day +4, maximum grade 3 by CTCAE and grade 3 by ASTCT[32], treated with steroids and anakinra). Prior to his diagnosis of COVID-19 and recurrent pneumonias, the patient did not have significant leukopenia/lymphopenia.

### Histopathology evaluation

Histopathologic analysis of the cervical LN biopsy was performed by hematopathologists at the Department of Pathology at the Brigham and Women's Hospital. H&E-stained and immunohistochemical stain slides of the cervical LN biopsy were prepared at the time of diagnostic workup at the Immunohistochemistry Laboratories in the Department of Pathology at the Brigham and Women's Hospital. A second core biopsy was obtained at the same time and flash frozen in Tissue-Tek O.C.T. Compound (Sakura) and cryopreserved until the time of spatial transcriptional analysis (see below). For TCL, the following antigens were evaluated: CD2, CD3, CD5, CD5, CD7, CD8, BLC6, Stathmin, PD-1, MUM1, BCL2, CD30, ALK-1, CD10, CD20, PAX5, CD79a, CD21, C-MYC, TCR-BF1, TCR-Delta, and Ki-67. In situ hybridization for EBV-encoded RNA was performed and found to be negative. Representative histopathological microphotographs were collected for publication.

### Bulk WGS

"Tumor" DNA was extracted from 12 sections of the remaining FFPE core LN biopsy. Genomic DNA was extracted from PBMC that were collected 12 months after CAR-T infusion. DNA was sequenced at the Broad Institute (Cambridge, MA), as previously described, at a depth of 60× for the "tumor" sample and 30× for the "normal" sample[33]. Bam files were input into our standard characterization pipeline, which includes MuTect for SNV calling[34], Strelka for indel calling[35], and HapASeg for copy number segmentation. DeTiN was used to rescue mutations observed at low frequency in the normal[36]. Additional filtering for artifactual mutations was performed using BLAT, a panel of normals, and filtering for oxidative damage[37].

Power to detect at least one read mapping to the axi-cel scFv was calculated under Poisson assumptions as (1):

$$power = P(nread > 0) = 1 - e^{-\lambda_{CAR}}$$

$$\lambda_{CAR} = \lambda_{autosomal} L r_{CAR} \rho$$

Where $\lambda_{autosomal} = 0.3$ is the number of read pairs per base observed in autosomal regions, L = 743 is the length of the scFv, $r_{CAR} = \frac{1}{2}$ is the copy ratio of the construct compared to autosomal regions assuming a single integration event, and $\rho$ is the fraction of cells with an integrated CAR construct.

### Rapid heme panel

Assessment of hematologic malignancy-associated DNA mutations was performed as part of routine clinical care at the time of the suspicious LN using the rapid heme panel developed at the Dana-Farber Cancer Institute and Brigham and Women's Hospital, an amplicon-based targeted next-generation sequencing platform designed to cover 95 genes in hotspot regions of oncogenes and coding regions of tumor suppressor genes[38]. The rapid heme panel was performed on DNA extracted from PBMCs.

### Slide-tags 5' snRNA-seq

Slide-tags were performed as previously described[25], with the below modifications for compatibility with 5' snRNA-seq. Capture sequence on spatial bead barcode oligonucleotides was replaced with the reverse complement of the 10× Genomics template switch oligonucleotide sequence (5'-CCCATATAAGAAA-3'). A 20 μm fresh-frozen tissue section from the LN biopsy was melted onto a 5.5 mm square bead array. The spatial tagging procedure and nuclei preparation were performed using our standard protocol. For the 5' snRNA-seq profile, 38.7 μl of counted nuclei was loaded into the 10× Genomics Chromium controller using the Chromium Next GEM Single Cell 5' Kit v2 (10× Genomics, PN-1000263). The Chromium Next GEM Single Cell 5' v2 Cell Surface Protein User Guide CG000330 Rev F was followed with small modifications. During step 2.2a, 1 μl of 0.3 μM spike-in primer (5'-GTGACTGGAGTTCAGACGT-3') was added and cDNA primers (PN-2000089) were used in place of Feature cDNA Primers 4 (PN-2000277). Spatial barcode indexing was performed similarly to step 6.1, but Dual Index Plate TT Set A was used in place of Dual Index Plate TN Set A. We performed 13 cycles of PCR according to step 6.1 conditions. A double sided SPRI (0.6× then 1.2×) was conducted for post sample index PCR size selection. We collected Slide-tags 5' snRNA-seq on two serial sections. The transcriptomic profiles consist of nuclei from both sections. Spatial positions are shown from the section with the most nuclei.

### Slide-tags data preprocessing

We used Cell Ranger (v.6.1.2)[39] mkfastq (10× Genomics) to generate demultiplexed FASTQ files from the raw sequencing reads. We aligned these reads to human GRCh38 and the Ensembl 100 annotation, supplemented with sequences for both the axi-cel and tisa-cel constructs[18] and quantified gene counts as UMIs using Cell Ranger count (10× Genomics). We used CellBender v.0.2.0 for background noise

correction and cell calling[40], setting --expected-cells to the number of Cell Ranger cell calls and --total-droplets-included to 40,000. Spatial data processing was performed as described in our original Slide-tags manuscript[25].

### T-cell gene expression program analysis

T-cell function is largely governed by the induction of gene expression programs. To explore the gene expression programs activated by the T-cells in our sample, we used T-CellAnnoTator (TCAT)[26]. TCAT uses nonnegative least squares regression to quantify the usage of a pre-defined set of context-diverse T-cell programs on our single-nucleus transcriptome data. We computed the mean usage scores of relevant programs for T-cell populations in the biopsied LN. We report the top 100 genes correlated with each program in Supplementary Data 3.

### T-cell receptor sequencing

Alpha and beta chain TCR sequences were enriched from the 5′ snRNA-seq cDNA libraries using the Chromium Single Cell Human TCR Amplification Kit (PN-1000252). TCR sequences were called using the cellranger vdj pipeline.

### Public TCR analysis

Human CDR3α and CDR3β sequences from public TCRs reactive to common viruses, including Influenza A, CMV, EBV, and SARS-CoV-2, were obtained from the VDJdb database (vdjdb.cdr3.net, accessed on 2024-09-11). Direct amino acid sequence matching of alpha and beta chains from the 5′ snRNA-seq data to the public TCRs was assessed.

### Viral reactivity signature scoring

T-cell transcriptomic profiles were scored on a set of genes associated with virus-specific T-cells using the AddModuleScore function in Seurat[41].

### Neighborhood enrichment permutation testing

To assess spatial neighborhood enrichment of pairwise cell type combinations, we used a permutation testing framework. For each center cell type, we calculated the average proportion of cells within a 200 μm radius belonging to the queried neighbor cell type. To establish a null distribution, we permuted cell type labels 1000 times and recomputed the average proportions for the queried neighbor cell type. A $Z$ score for the cell type combination was then computed by comparing the observed average proportion to the null distribution. This procedure was repeated for each pairwise cell type combination.

### TET2 mutation identification from snRNA-seq reads

*TET2* p.776Wfs*4 mutant reads were counted from the snRNA-seq BAM file and the consensus genotype of each UMI was determined. Cell barcodes with at least one variant-containing UMI were considered mutant. Cell barcodes with no variant-containing UMIs but at least one UMI spanning the queried sequence position were considered wild-type. All other cell barcodes were not assigned a genotype.

### qPCR for the CAR transcript

The 5′ snRNA-seq cDNA library was obtained for qPCR to detect the CAR transcript. cDNA was synthesized from RNA isolated from an infusion product bag washing from a patient who received CAR-T therapy[42] as a positive control and from healthy donor T cells as a negative control. Primer sets were designed to target FMC63[43] to detect the CAR transcript and the housekeeping gene *GAPDH* (ThermoFisher). qPCR was performed as previously described[44].

### Reporting summary

Further information on research design is available in the Nature Portfolio Reporting Summary linked to this article.

## Data availability

Single cell transcriptome, TCR, and WGS data will be submitted to NCBI's Database of Genotypes and Phenotype (dbGaP; https://www.ncbi.nlm.nih.gov/gap) under accession code phs002922.v3.p1. Source data are provided with this paper.

## Code availability

Code for processing spatial sequencing libraries is available at GitHub (https://github.com/broadchenf/Slide-tags).

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

## Acknowledgements

We thank Doreen Hearsey, Allison Anderson and all staff from the Ted and Eileen Pasquarello Tissue Bank in Hematologic Malignancies for excellent technical support with banking of clinical samples. C.J.W. is a member of the Parker Institute for Cancer Immunotherapy at DFCI. Her work is supported, in part, by the Parker Institute for Cancer Immunotherapy. C.J.W. is also the Lavine Family Chair for Preventive Cancer Therapies at DFCI. C.J.W. and F.C. are supported by The Mark Foundation. K.M. is supported by the Lubin Family Foundation Scholar Award. F.C. acknowledges funding from the NCI (R01 CA276865, R33CA291199) and the NYSCF. F.C. is an NYSCF Roberston Investigator. We thank the patients for generous contribution of clinical data and research samples for this study. We are thankful to T. Coorens, V. Shanmugam, C.K. Hahn, D. Fisher, P. Armand, and R.J. Soiffer for helpful discussions and feedback.

## Author contributions

K.M., J.A.W., C.J.W., F.C., and C.J. conceived and designed the study. K.M., H.B., M.M., L.L., and J.D.C. identified and obtained patient samples and relevant clinical information. K.M., H.B., G.P., B.W., P.D.M., J.P.L., and C.J. interpreted patient data. K.M., J.A.W., E.L., and A.N. performed experiments. K.M., J.A.W., N.J.H., L.G., Q.G., M.B., and G.G. designed, performed, and interpreted data analysis. J.A.W., J.S., S.A., and V.K. developed Slide-tags 5′ snRNA-seq. All authors participated in manuscript writing and review and provided final approval of the manuscript.

## Competing interests

C.J. reports consultancy for Kite/Gilead, BMS/Celgene, Novartis, Instil Bio, ImmPACT Bio, Caribou Bio, Miltenyi, Ipsen, ADC Therapeutics, Abbvie, AstraZeneca, Morphosys, Synthekine, and Sana, and research funding from Kite/Gilead and Pfizer. C.J.W. is an equity holder of BioNTech, Inc., receives research funding from Pharmacyclics, and is on the SAB of Repertoire, Adventris, Nature's Toolbox, Aethon Therapeutics. F.C. is an academic founder of Curio Bioscience and Doppler Biosciences, and a scientific advisor for Amber Bio. F.C.'s interests were reviewed and managed by the Broad Institute in accordance with their conflict-of-interest policies. The remaining authors declare no competing interests.
