## [Transparent Peer Review file · Nature Communications]

A clonally expanded nodal T-cell population diagnosed as T-cell lymphoma after CAR-T therapy

Corresponding Author: Dr Caron Jacobson

Version 0:

Reviewer comments:

Reviewer #1

(Remarks to the Author)

Dr. Maurer and colleagues re-submit an interesting case of a CAR-negative T-cell lymphoproliferation. This topic continues to be of interest and there are three recently reported CAR+ neoplasms in NEJM after cilta-cel published in the interim (Harrison et al, NEJM 2025, Perica et al, NEJM 2025).

The authors revised draft includes qPCR analysis of scRNA cDNA for more sensitive detection of the CAR vector, additional analysis of their previously generated spatial data, and additional discussion including of functional variants discovered in the T-cell lymphoproliferation.

This case combines with other reports of CAR-negative T-cell lymphoproliferations in CD19/CD28 vectors offering additional insight into a field of high importance but with limited total reports.

The authors definitively show the lymphoproliferation is CAR-negative. The entity remains unclear in its characterization. The authors make the reasonable point that caution is warranted with such neoplasms, and it is true that post-CAR patients have highly clonal native T-cell populations that could be mistaken for a neoplasm (see Strati et al, Cell rep med). The author's methodology is sound and the conclusions reasonable.

Some opportunities are missed in the report including deeper sequencing of a leukapheresis sample and sample taken at 1 year prior to the lymphoproliferation which could have provided clonal insights into the entity. That said the authors have made a good effort at addressing all reviewer comments including generating new data and including new analysis/discussion. The report is appropriately impactful for the journal and the language sufficiently cautious surrounding characterization of the entity.

Reviewer #2

(Remarks to the Author)

Overall, this updated manuscript is strengthened by the added spatial transcriptomic analyses and expanded genetic characterization. Given the manuscript's expanded data and refined discussion, it is a strong candidate for publication in Nature Communications, as it provides valuable insights into distinguishing malignant TCL from reactive T-cell proliferations.
